# Treatment Classification by Intent in Oncology—The Need for Meaningful Definitions: Curative, Palliative and Potentially Life-Prolonging

**DOI:** 10.3390/jpm14090932

**Published:** 2024-08-31

**Authors:** Zsolt Fekete, Andrea Fekete, Gabriel Kacsó

**Affiliations:** 1Department of Oncology, “Iuliu Hațieganu” University of Medicine and Pharmacy, 400015 Cluj-Napoca, Romania; gabriel.kacso@umfcluj.ro; 2“Prof. Dr. I. Chiricuță” Oncology Institute, 400012 Cluj-Napoca, Romania; 3Care and Assistance Center for Adults with Disabilities, 400304 Cluj-Napoca, Romania; csspad@dgaspc-cluj.ro; 4Amethyst Radiotherapy Center, 407280 Floresti, Romania

**Keywords:** treatment goal, curative treatment, palliative treatment, potentially life prolonging treatment, supportive care, personalized approach

## Abstract

Background: Realistic cancer treatment goals should be used by health care professionals and communicated to patients, families, and the public. The current nomenclature on this subject is outdated and has not been changed since the advent of modern oncology in the middle of the 20th century. Methods: Based on the literature we propose a three-tier system composed of curative, palliative, and potentially life-prolonging (PLP) therapies, instead of the current two-tier system of only curative and palliative treatment. Results: The new system introduces the notion of prolonged survival. Furthermore, the negative connotation linked to palliative care is also eliminated in this setting. Conclusion: The current terminology used to describe cancer treatment goals has not been updated since the mid-20th century and it is time for a more modern approach. We propose a three-tier system: (1) curative treatment, (2) palliative care, and (3) potentially life-prolonging therapy.

## 1. Introduction

In oncology, classically, management can be broadly categorized into two main approaches: curative and palliative. Historically, the management of cancer was only palliative, with rare cases of cures, for example, the resection of incipient breast cancer or skin basocellular carcinoma by early surgeons. Sometimes, prolongation of survival could be achieved by local treatments which offered short-term control and occasionally not only palliation, but longer life as well, even if by a small amount [1]. The introduction of radical oncological surgery and radiotherapy offered a cure for a much higher number of patients. Chemotherapy can rarely cure cancer, especially solid tumors, even today. Certainly, hematologic malignancies and germ cell tumors can be cured in a considerable proportion by chemotherapy, but these are mostly the exceptions from the general rule. Chemotherapy, when first introduced, was used mainly to palliate disease symptoms of inoperable cases, while later the prolongation of survival and curing was possible, in adjuvant or neoadjuvant settings, in combination with surgery and/or radiation therapy [2]. Other treatment modalities, such as hormone therapy, molecular targeted treatments such as kinase-inhibitors and immune therapy, high-intensity focalized ultrasound, local and general hyperthermia, cryotherapy, organ and tissue transplantation, the administration of anti-bodies carrying radioactive elements, dietary interventions and supplements, complementary methods (plant and fungal extracts), etc., can all either cure or induce the prolongation of survival or palliation [3].

Realistic cancer treatment goals should be used by health care professionals and communicated to patients, families, and the public. The current nomenclature on this subject is outdated and has not been changed since the advent of modern oncology in the middle of the 20th century. Based on the literature, we propose a three-tier system composed of curative, palliative, and potentially life-extending therapies, instead of the current two-tier system of only curative and palliative treatment.

## 2. The Current Two-Tier System: Curative and Palliative Treatment

The National Cancer Institute from the United States defines palliative care as follows: “*…to improve* the quality of life of *patients (…)*” [4].

It states that “*it can be given with or without curative care*”, and thus curative and palliative treatments can overlap. Furthermore, “*palliative care is an approach to care that addresses the person as a whole, not just their disease*”.


*“The goal is to prevent or treat, as early as possible, the symptoms and side effects of the disease and its treatment, in addition to any related psychological, social, and spiritual problems”.*


Cancer Research UK states [5] that “*the aim of palliative treatment is to relieve symptoms and improve quality of life*”.


*“It can be used at any stage of an illness if there are troubling symptoms, such as pain or sickness. In advanced cancer, palliative treatment might help someone to live longer and more comfortably, even if they cannot be cured”.*


Cancer Research UK lists treatment types that can palliate cancer: chemotherapy and other medicines (such as painkillers), radiotherapy, hormone therapy, targeted cancer drugs, surgery, radiofrequency ablation and cryotherapy.

We can observe that palliative treatment can include the achievement of prolonged survival as well.

The synonyms used for curative treatment are “curative intent” or “radical therapy/intervention”. The synonym for palliative treatment is “supportive care”, although the latter can have a broader meaning [6].

End of life care is a subtype of supportive care, when the patient is expected to succumb to their illness in a shorter period of time, usually less than a month, or patients have a performance index of 4. The European Society of Medical Oncology (ESMO) defines end-of-life care as “care for people with advanced disease once they have reached a point of rapid physical decline, typically the last few weeks or months before an inevitable death as a natural result of a disease” [7]. The NHS UK defines “end of life care” as “support for people who are in the last months or years of their life” [8].

Acute oncology targets conditions and situations which radically shorten the patients’ survival or induce severe complications, such as paralysis or organ damage. Its intent can be curative (chemotherapy in leukemia), palliative (treatment of spinal cord injury), or life-extending, but not curative (treatment of superior vena cava obstruction) [9].

The term ‘palliation’ now carries a negative implication as well, suggesting that the patient is in the final stages of his treatment.

## 3. The Proposed Three-Tier System: Curative, Potentially Life-Prolonging (PLP), and Palliative

We analyzed the fundamentals of the current two-tier division of treatment goals for cancer, i.e., curative vs. palliative and how it fits in our daily practice as radiation or medical oncologists or clinical psychologists. We propose to introduce the term “potentially life-prolonging therapy” (PLP) to subdivide palliative care into (1) true palliation, which aims only at symptom reduction, and (2) a treatment approach with potential for extending survival. For curative intent treatment we suggest also a dichotomy into (3) true curative (when a large proportion of patients are free of disease at a certain time point long enough after anticancer therapy has ended) and (4) all the remainders currently named curative, but actually receiving a potential life-prolonging (PLP) therapy (as for glioblastoma). Subgroups (2) and (4) will merge in the PLP newer entity. What the magnitude of the large versus the small proportion is to allow classification into (3) vs. (4) is unknown and ethically highly debatable. The same applies for the measurement of the cure rate with the classical surrogate of “alive and well with no evidence of disease at 5 years from the treatment”.

We also propose that a reference time point for a “curative goal” should be related to a patient’s natural life expectancy and to the biology of the disease, instead of the ubiquitarian 5-year overall survival. That might reclassify, for example, a radical prostatectomy or a definitive radiotherapy for localized prostate cancer (mainly low-risk) as just a PLP (according to PIVOT or ProtecT randomized trials) [10,11] or even a palliative approach for a urinary symptomatic patient with a natural life expectancy of less than 10 years (related to his age, co-morbidities and national averages).

Curative or PLP therapy can also have a palliative role, such as having direct action on alleviating or the complete remission of symptoms related to cancer, like obstruction, bleeding, or pain. In this respect, palliative therapy is most correctly named palliative-only treatment.

According to the proposed classification, surgery can be curative (intent)/radical, palliative, or potentially life-prolonging (for example, surgery in a colon obstruction). Radiotherapy is mostly either curative or palliative, although there is some middle ground of potentially life-extending treatment, as stated in Table 1. Chemotherapy is mostly “potentially life-prolonging”, rarely curative, at least alone, and palliative. The combination of these treatments can also be named after the three criteria: neoadjuvant chemotherapy with s curative intent, followed by surgery; curative chemoradiation; potentially life-extending chemoradiation (for example, in locally advanced, bulky esophageal cancer).

The proposed classification system is presented in Table 1.

We recognize that even if this proposal is accepted by the medical community, it will take some time to implement. Although the difference is minimal, it is perceived as an unmet need by many patients, caregivers, the pharmaceutical industry (which sponsors clinical trials), as well as health technology assessment entities and insurance companies, both public and private. To improve compliance, a transitional period of a few years could be beneficial, similar to the shift from the 6-to-10 Gleason score to the 1-to-5 ISUP/WHO 2016 grading system for prostate cancer. We believe that associating the three-layer therapy goal with letters and/or colors, like traffic lights, will facilitate standardization and acceptance, especially for patients.

## 4. Current and Future Perspectives

Researchers have previously acknowledged the significance of precisely defining treatment goals. Markman [12] defined the following possible goals: cure, the prolongation of survival, improvement in quality of life, the palliation of symptoms, and the prevention of complications. By our three-tier definition, we try to simplify the Markman model but also add clarity to the intermediate term between curative and pure palliative. Why is our general proposed term “potentially life-*prolonging*” (PLP) and not “life-extending” as proposed in 2017 by Neugut et al. [13]? An intervention can have no effect on the survival in an individual patient, even when the average survival values are better in a patient cohort. Furthermore, an intervention can even shorten survival in a certain subject through a side effect, such us febrile neutropenia in the case of chemotherapy, infection, or hemorrhage in the case of surgery and “hyper-progression” or autoimmune conditions induced by immune-checkpoint inhibitors, just to name a few.

To declare a subject cured, one typically needs to wait approximately 5 years for most cancers, possibly less (around 3–4 years) for lymphomas [14], or significantly longer (10 years at least) for prostate and breast cancers, with “no evidence of disease” status by acknowledged clinical, imaging and tumor marker assessments [15]. Thus, the match of expectations and treatment results will mature after several years. Circulating tumor DNA (ctDNA) can be used potentially in the future to confirm with a higher probability if a patient needs adjuvant treatment or can reassure that the treatment indeed was a radical one. Thus, ctDNA could be used in the future as a marker of minimal residual disease also for solid tumors, as it is used for some hematological malignancies; the clinical investigations are ongoing [16,17].

Naturally, treatment goals might differ from treatment results. For example, even though the treatment goal in stage III, inoperable, non-small cell lung cancer (NSCLC) is to cure it, with the aid of chemoradiation, the 5-year overall survival (5yOS) as a surrogate for the cure rate is only around 20–30% [18]. For the remaining 70–80% of patients, only the prolongation of survival and/or the palliation of symptoms (dyspnea, hemoptysis) can be achieved, and, in less than 5%, a treatment-related death might occur due to neutropenia-related infections or non-neutropenic pneumonia. For PD-L1 > 1% stage III NSCLS not progressing under/immediately after chemoradiation, the use of adjuvant Durvalumab for 1 year significantly increases (by 10–15%) the curative proportion and the 5yOS exceeded the >50% threshold for fit time [19]. This is a clear example of how a biomarker-driven therapy (the PDL1 status) can convert a curative yield from a minority into a majority. In other words, if we simply set the bar at 50% between curative and potentially life-extending treatment (PLP), the chemoradiation (CRT) for this selected group of patients would be considered a Y-code, whereas CRT+ durvalumab would be a G-code by our proposed system (Table 1).

Obviously, the difference in a minority (<50%) versus a majority (>50%) as an argument for different coding is an oversimplification. It does not apply for cancers where the outcome is very good with surgery alone (5yOS > 90%), for example, endometrial endometroid stage I carcinoma and where the adjuvant brachytherapy or external beam radiotherapy based on risk factors (age > 60 years, lympho-vascular space involvement, the grade and depth of myometrial invasion) marginally improves the local or locoregional control without a 5yOS benefit. This lack of survival does not make radiation therapy a palliative treatment. The genomic ProMisE classification [20] as a biomarker allowed a more personalized adjuvant approach for the same stage I endometrial carcinoma after surgery, and it was included in the latest 2023 FIGO staging. It avoids both overtreatment (no need for RT in the POLE-mutated subgroup, 6–7%) or undertreatment (the need for adjuvant chemotherapy and external beam RT for the p53-mutated cancers) and allows for the potential replacement of the classical adjuvant armamentarium (RT and or chemotherapy) with immunotherapy for microsatellite instability (MSI) or mismatch repair-deficient tumors (MMRd, 25–30% of patients), as it was recently been proven for stage III–IV [21]. HER2-overexpressed or homologous recombination-deficient recurrent or primary stage III–IV endometrial cancers are eligible for targeted therapy such as Trastuzumab-Deruxtecan or PARPi in different scenarios, including maintenance therapy as a PLP strategy in our view [22,23] as significant better median survival exists besides symptom relief, but with curing being very unlikely. 

In cases of glioblastoma, following macroscopic tumor removal, adjuvant chemoradiation is administered with a curative intent. However, the cure rate remains below 5%. For the remaining patients, it may lead to a prolongation of survival [24]. The hypermethylated MGMT-promoter status as a biomarker nearly doubles the cure rate, but it remains quite low, at less than 10%. In this clinical setting, the treatment could be defined as a PLP (Y-code), a potentially life-prolonging intervention.

The subdivision of palliative care is especially important when explaining clinical trial goals to a potential participant and to the patient’s family. The use and benefit of purely palliative chemotherapy is highly debated, although patient preference can be considered in this setting. Research on chemotherapeutic palliation, assessed through valid quality-of-life measures, reveals that patients may be willing to tolerate certain treatment side effects if they experience relief from tumor-related symptoms [25]. Often, patients who receive palliative chemotherapy have been shown to have false expectations. In the study of Wright et al. [25], patients receiving end-of-life chemotherapy were also more likely to express a preference to receive “life prolonging” care over comfort care (39% vs. 26%, *p* = 0.01), including chemotherapy if it might extend their life by one week (86% vs. 60%, *p* < 0.001), compared with patients not receiving end-of-life chemotherapy. An acceptable toxicity trade-off for an additional week of life would be difficult if not impossible to measure. 

Patients diagnosed with incurable cancer often confront difficult decisions regarding their treatment options. These choices involve weighing the possibility of extending their life (length of life, or LL) against the impact on their overall quality of life (QoL). However, little information exists about patients’ preferences and attitudes toward these trade-offs. A recent review by Shrestha and colleagues [26] aims to address this gap by exploring the complex factors that influence patients’ decisions when choosing between their QoL and LL. According to the authors, patients often prioritize survival when making treatment decisions, but the current health status of the patients also affected their choice: subjects in better health were found to rate their LL more highly, whereas those who were in poorer health strived to maintain their QoL. There are demographic factors influencing treatment decisions as well. For example, subjects with strong family links prefer survival and unemployed patients prioritized QoL in larger numbers than those currently employed.

Often, the expectations of patients with advanced cancer are unrealistic and “un-informed”. Mohammed et al. [27] identified seven types of patients with advanced cancer who pursue potentially life-extending cancer treatments, when they do not exist: (1) the desperate, (2) the cancer expert, (3) the proactive, (4) the productive, (5) the mistrusting, (6) the model patient, and (7) the suffering subject. All these subjectivities are maladaptations to the diagnosis of uncurable cancer, when life-extending therapies do not exist. The authors define the concept of “conflicted dying” as patients “simultaneously having life-threatening cancer and actively searching for life extension”. All these seven subcategories are purely palliative, not PLP.

The adoption of the terms “palliative” and “potentially life-prolonging treatment” is specifically difficult for radiotherapy since clinical trials with proven survival benefits are much less numerous than those for systemic treatments. Moreover, a significantly better loco-regional control provided by a certain RT protocol does not necessarily translate into a better cure rate because of metastases’ onset as a failure not related per se to the delivered radiotherapy. On the other hand, from a broader perspective, the PLP subdivision suits perfectly stereotactic ablative radiotherapy for oligometastatic subsets [28].

Our proposed three-tier system, integrating the new PLP subset, is certainly not perfect, but it has the merit to better adjust to modern personalized oncology, where molecular classifications and new biomarker discovery increase our ability for an improved, tailored treatment approach beyond the (too) simple curative–palliative binomial taxonomy.

## 5. Conclusions

Labeling all non-curative cancer treatment options as palliative is inaccurate and outdated. We should consider adopting a novel semantic framework to more accurately describe the survival advantages resulting from various non-curative treatment choices.

The proposed three-tier system can be extended to all areas of medicine, not just oncology. It provides a more comprehensive framework for treatment goals, emphasizing curative, palliative, and potentially life-prolonging (PLP) approaches, with significant implications not only for the patient’s better understanding of the proposed treatment strategy, but also for the coding, health technology assessment, and reimbursement from health insurance.

## Figures and Tables

**Table 1 jpm-14-00932-t001:** Treatment types in oncology by goals.

Term	Intent and Goals	Examples
Curative treatment G code Green color	The intent is to cure the tumor. Tumor removal is performed while adhering to the principles of oncological surgery. Alternatively, it can include non-surgical curative-intent treatment modalities. Organ preservation with a curative goal is included.	Surgery: radical surgery for localized prostate cancer, lung cancer, etc.
		Radiotherapy: radiotherapy of head and neck cancer with a curative intent.
		Chemotherapy/modern systemic treatment: chemotherapy for testicular cancer; adjuvant, neoadjuvant, induction or concomitant chemotherapy in different curative approaches (in particular, as organ preservation strategies for advanced pharyngo-laryngeal carcinoma, bladder or lower rectal cancer).
Potentially life-prolonging (PLP) treatment Y code Yellow color	The treatment applied was proven to prolong survival in previously published clinical studies (all phases of clinical studies and retrospective cohorts). Palliation of symptoms can occur. Cancer is transformed into a chronic disease; cure is very unlikely, although it can occur extremely rarely.	Surgery: removal of kidney primary or colorectal primary in metastatic cancer.
		Radiotherapy: radiotherapy for locally advanced, inoperable breast cancer; radiotherapy to the primary for oligometastatic prostate cancer;
		chemotherapy/modern systemic treatment: immune checkpoint inhibitors in metastatic NSCLC.
Palliative (-only) treatment R-code Red color	The scope of the treatment is to reduce the patient’s suffering (symptom relief) WITH NO proven survival benefit.	Surgery: surgical nerve block.
		Radiotherapy: radiotherapy for painful bone metastases.
		Chemotherapy/modern systemic treatment: third-line chemotherapy in metastatic lung cancer.

## Data Availability

No new data were created or analyzed in this study. Data sharing is not applicable to this article.

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
