# Peer review of "Treatment Classification by Intent in Oncology—The Need for Meaningful Definitions: Curative, Palliative and Potentially Life-Prolonging"

_jpm, 2024, doi:10.3390/jpm14090932_

Round 1

Reviewer 1 Report

Comments and Suggestions for Authors

the issue is an important topic as palliative an palliative are not always the same. 

"Labelling all non-curative cancer treatment options as palliative is inaccurate and outdated." I can identifiy with this statement but I find it difficult to talk about patient-oriented concepts here, but then to deal with complex terminology. "Potentially life-prolonging therapy" is a term that absolutely not every patient understands immediately. Education is the key to patient-oriented care and I would like the authors to explain how they intend to implement and use the terminology. A terminology that has been established decades ago is very hard to alter.  On the other hand I am convinced that an excellent health carer is capable of explaining to the patient what he or she can expect from the current therapy independently from nomenclature. 

If the authors suggest a new terminology here, I think it would be appropriate to also work out the implementation (e.g. color code, information flyer, etc.). Otherwise there is no benefit for the patient and the health care providers will be hard to convince to change a decades-old nomenclature.

Author Response

Dear Colleague,

We thank you for the time and effort of reviewing our work.

We appreciate your valuable feedback and constructive criticism.

We tried to address them as follows:

OBS 1: Related to your review statement: "Labelling all non-curative cancer treatment options as palliative is inaccurate and outdated." I can identify with this statement, but I find it difficult to talk about patient-oriented concepts here, but then to deal with complex terminology. "Potentially life-prolonging therapy" is a term that absolutely not every patient understands immediately

               We agree with your  opinion that potentially life prolonging therapy (PLP)  is a rather complex new terminology, but we believe it capture at its best the nuances of this intermediate setting between curative and palliative. We would be happy to accept a simpler term from your part but we did not manage to find a better one. From this perspective, we think that neither “palliative” is not a concept easy to grasp for every patient. Yet , we are still all using it, because - as you clearly said afterwards- - we are “ explaining to the patient what he or she can expect from the current therapy” by additional words to translate the abstract concepts.

OBS2: related to your advise “ it would be appropriate to also work out the implementation (e.g. color code, information flyer, etc.)”.

               We gladly incorporated your excellent suggestion  into the table 1 and in the text, adding alternatively capital letters code to the traffic light colors concept  (row130-132): We feel that pairing the 3 layer therapy goal with letters and/or colors like traffic lights will ease the standardization and acceptance, at least for the patients and insurances

Table 1. Treatment types in oncology by goals.

Term

Intent and goals

Examples

Curative treatment

G code

Green color

The intent is to cure the tumor. Tumor removal is performed while adhering to the principles of oncological surgery. Alternatively, it can include non-surgical curative-intent treatment modalities.

Organ preservation with curative goal is included

Surgery: radical surgery for localized prostate cancer, lung cancer etc.

Radiotherapy: curative intent radiotherapy of head and neck cancer

Chemotherapy/modern systemic treatment: chemotherapy for testicular cancer; adjuvant, neoadjuvant, induction or concomitant chemotherapy in different curative approaches (in particular as organ preservation strategies such advanced pharyngo-laryngeal carcinoma, bladder or lower rectal cancer)

Potentially life-prolonging (PLP) treatment

Y code

Yellow color

The treatment applied was proven to prolong survival in previously published clinical studies (all phases of clinical studies and retrospective cohorts). Palliation of symptoms can occur.

Cancer is transformed into a chronic disease; but cure is very unlikely.

Surgery: removal of kidney primary or colorectal primary in metastatic cancer

Radiotherapy: radiotherapy for locally advanced, inoperable breast cancer; radiotherapy to the primary for oligometastatic prostate cancer

Chemotherapy/modern systemic treatment: immune checkpoint inhibitors in metastatic NSCLC

Palliative(-only) treatment

R -code

Red color

. The scope of the treatment is to reduce patient’ suffering (symptom relief) WITH NO proven survival benefit.

Surgery: surgical nerve block

Radiotherapy: radiotherapy for painful bone metastases

Chemotherapy/modern systemic treatment: third line chemotherapy in metastatic lung cancer

Obs3:  related to your comment’ the health care providers will be hard to convince to change a decades-old nomenclature:

àwe added a suggestion of ’possible better adherence to this new PLP concept in the paragraph 124-130: We are aware that even if accepted by the medical community, it will take some time for such a switch, although the difference is minimal, but felt as an unmet need for many patients, some care givers, pharma industry (sponsoring clinical trials) but also for health technology assessment entities or insurance companies, whether public or private. A possible solution for a higher compliance would be a transitory period of a couple of years, like the1 to 5 ISUP/WHO 2016 grading system for prostate cancer replacing the previous 6 to 10 Gleason score.

We hope that we were able to properly adjust in accordance with your insights related to our proposed paper.

Sincerely yours,

The authors

Reviewer 2 Report

Comments and Suggestions for Authors

In this paper, Fekete et al. discuss about the need to establish a 3-tier system regarding treatment goals when caring for oncology patients. The authors argue that the adding a more defined category ("potentially life-extending") can be more informative to the patients and set clear expectations. While the authors make a decent argument, I do not think that the paper (in its current form) represents a significant contribution to current literature. I believe that the paper represents more of a superficial "opinion piece/letter", rather than an original study. I think the impact of this work could be significantly improved if the authors focus on providing a more in-depth analysis of their own thoughts, backed up by published literature, rather than adding a lot of information that are not really relevant. For example, lines 131-161 are not necessary - or I can't see how they fit with the rest of the manuscript. 

Comments on the Quality of English Language

Moderate editing needed. 

Author Response

Dear Colleague,

We thank you for the time and effort of reviewing our work.

We appreciate your valuable feedback and constructive criticism.

We tried to address them as follows

OBS1. “The authors argue that adding a more defined category ("potentially life-extending") can be more informative to the patients and set clear expectations. While the authors make a decent argument, I do not think that the paper (in its current form) represents a significant contribution to current literature. I believe that the paper represents more of a superficial "opinion piece/letter", rather than an original study.

  • We understand your concern, and we agree that our article is not a genuine original study, but rather a review article on an important concept definition on treatment goals in oncology. We believe that the original part is not as much related to the new “Potentially life prolonging (PLP)” category, but to its ‘actual need in our daily practice, from regular consultation to clinical trials, and its implications for patients, care givers and health system, balancing resources against better defined treatment goals. We foster that angle by restructuring (hopefully clearer, with stronger arguments) with add-ons the 3-tier system (lines 88-145, and table 1)

OBS 2. I think the impact of this work could be significantly improved if the authors focus on providing a more in-depth analysis of their own thoughts, backed up by published literature,

               We followed your advice and enhanced the applicability of the proposed model, underlying also its’ limits, by adding/ restructuring lines 157-197, plus new references.

Naturally, treatment goals might differ from treatment results. For example, even though the treatment goal in stage III, inoperable non-small cell lung cancer (NSCLC) is cure, with the aid of chemoradiation, the 5 year overall survival (5yOS) as surrogate for cure rate is only around 20-30%. [18] For the remaining 70-80% of patients only prolongation of survival and/or palliation of symptoms (dyspnea, hemoptysis) can be achieved, and, in less than 5%, a treatment related death might occur due to neutropenia-related infections or non-neutropenic pneumonia. For PD-L1 >1 % stage III NSCLS not progressing under/immediate after chemoradiation, adjuvant Durvalumab for 1 year significantly increases (by 10-15%) the curative proportion and the 5y OS exceeded for the fits time the >50% threshold [19]. This is a clear example of how a biomarker driven therapy (the PDL1 status) can convert a curative yield from minority into majority. In other words, if we simply set the bar at 50 % between curative and potentially -life-extending treatment (PLP) the chemoradiation (CRT) for this selected group of patients would be consider Y-code, whereas CRT + durvalumab would be a G-code by our proposed system (table 1).

Obviously, the difference from minority (<50%) versus majority (>50%) as argument for different coding is an oversimplification. It does not apply for cancers where the outcome is very good with surgery alone (5yOS >90%) for example endometrial endometroid stage I carcinoma and where the adjuvant brachytherapy or external beam radiotherapy based on risk factors (age>60 years, lympho-vascular space involvement, grade and depth of myometrial invasion) improves marginally the local or locoregional control without 5yOS benefit. This lack of survival does not make radiation therapy a palliative treatment. The genomic ProMisE classification [20] as biomarker allowed a more personalized adjuvant approach for the same stage I endometrial carcinoma after surgery, which was included in the latest 2023 FIGO staging  It avoids both overtreatment (no need for RT in the POLE mutated subgroup, 6-7%) or undertreatment (need for adjuvant chemotherapy and external beam RT for the p53 abnormal cancers) and allows potential replacement of the classical adjuvant armamentarium (RT and or chemotherapy) by immunotherapy for microsatellite instability (MSI) or mismatch repair deficient (MMRd, 25-30% of patients), as it was recently been proven for stage III-IV [21]. HER2 overexpressed or homologous recombination. deficient recurrent or primary stage III-IV endometrial cancers are eligible for targeted therapy such Trastuzumab-Deruxtecan or PARPi in different scenarios, including maintenance therapy as PLP strategy in our view [22,23] as significant better median survival besides symptom relief but with cure very unlikely.

In cases of glioblastoma, following macroscopic tumor removal, adjuvant chemoradiation is administered with a curative intent. However, the cure rate remains below 5%. For the remaining patients, it may lead to a prolongation of survival. [24] The hypermethylated MGMT-promoter status as biomarker improves the cure odd by almost doubling it, but unfortunately remaining still very low (less than 10 %). In these clinical settings the treatment could be defined as a PLP (Y -code), potentially life-prolonging intervention.

    …..and  lines235-246:

The adoption of the terms “palliative” and “potentially life-prolonging treatment” is specifically difficult for radiotherapy since clinical trials with proven survival benefit are much less numerous than for systemic treatments. Moreover, a significant better loco-regional control provided by a certain RT protocol does not translate necessarily into a better cure rate, because of metastases onset as failure not related per se to the delivered radiotherapy On the other hand, from a broader perspective, the PLP subdivision suits perfectly stereotactic ablative radiotherapy for oligometastatic subsets.[28].

Our proposed 3-tiers system, integrating the new PLP subset, is certainly not perfect but it has the merit to better adjust to the modern personalized oncology, where molecular classifications and new biomarker discovery increase our ability for an improved tailored treatment approach beyond the (too) simple curative-palliative binomial taxonomy.

Obs3. “rather than adding a lot of information that are not really relevant. For example, lines 131-161 are not necessary - or I can't see how they fit with the rest of the manuscript. 

  • We underscored upfront the limits of the current 2 choice system Curative/Palliative in particular for patients’ expectations, mainly in metastatic setting. On the other hand we do not advocate for a too complicated alternative, without relevance as you truly pointed out. We restricted that part and kept only the paragraphs of others extensive previous work that serve as argument for our 3-tiers system as simpler than their sophisticated analysis for pure palliative scenario: .

The subdivision of palliative care is especially important when explaining clinical trial goals to a potential participant and to the patient’s’ family. The use and benefit of purely palliative chemotherapy is highly debated, although patient preference can be considered in this setting. Research on chemotherapeutic palliation, assessed through valid quality-of-life measures, reveals that patients may be willing to tolerate certain treatment side effects if they experience relief from tumor-related symptoms. [25] Often, patients who receive palliative chemotherapy have been shown to have false expectations. In the study of Wright et al [25] patients receiving end-of- life chemotherapy were also more likely to express a preference to receive “life prolonging” care over comfort care (39% vs 26%, P=0.01), including chemotherapy if it might extend their life by one week (86% vs 60%, P<0.001), compared with to patients not receiving end-of-life chemotherapy. What is the trade-off an acceptable toxicity for an additional week of life would be difficult if not impossible to measure. Patients diagnosed with uncurable cancer often confront difficult decisions regarding their treatment options. These choices involve weighing the possibility of extending their life (length of life, or LL) against the impact on their overall quality of life (QoL). However, little information exists about patients’ preferences and attitudes toward these trade-offs. A recent review by Shrestha and colleagues [26] aims to address this gap by exploring the complex factors that influence patients’ decisions when choosing between QoL and LL. According to the authors, patients often prioritize survival when making treatment decisions, but the current health status of the patients also affected their choice: subjects in better health were found to rate LL more highly, whereas those who were in poorer health strived to maintain their QoL. There are demographic factors influencing treatment decisions as well. For example, subjects with strong family links prefer survival and unemployed patients prioritized QoL in larger numbers than those currently employed

Often the expectations of patients with advanced cancer are unrealistic and “un-informed.” Mohammed et al. [27] identified seven types of patients with advanced cancer who pursue potentially life-extending cancer treatments, when they do not exist: (1) the desperate, (2) the cancer expert, (3) the proactive, (4) the productive, (5) the mistrusting, (6) the model patient, and (7) the suffering subject. All these subjectivities are maladaptation to the diagnosis of uncurable cancer, when life-extending therapies do not exist. The authors define the concept of “conflicted dying” as patients “simultaneously having life-threatening cancer and actively searching for life extension.” All these 7 subcategories are purely palliative, not PLP.

We hope that we were able to properly adjust in accordance with your insights related to our proposed paper.

Sincerely yours,

The authors

Round 2

Reviewer 2 Report

Comments and Suggestions for Authors

I would like to thank the authors for providing a revised version of their manuscript. I believe that the extensive additions and changes have made their paper worth reading. 

Comments on the Quality of English Language

Minor grammatical errors and typos. Please re-examine before final submission.

Author Response

Dear Colleague,

thank you again for your meticulous review. We have checked the paper once more for typos and grammar issues.

Sincerely,

the Authors